# Leucine Supplementation Exacerbates Morbidity in Male but Not Female Mice with Colorectal Cancer-Induced Cachexia

**DOI:** 10.3390/nu15214570

**Published:** 2023-10-27

**Authors:** Eleanor R. Schrems, Wesley S. Haynie, Richard A. Perry, Francielly Morena, Ana Regina Cabrera, Megan E. Rosa-Caldwell, Nicholas P. Greene, Tyrone A. Washington

**Affiliations:** 1Exercise Muscle Biology Laboratory, Department of Health, Human Performance and Recreation, University of Arkansas, 155 Stadium Dr. HPER 309, Fayetteville, AR 72701, USA; erschrem@uark.edu (E.R.S.);; 2Cachexia Research Laboratory, Exercise Science Research Center, Department of Health, Human Performance and Recreation, University of Arkansas, Fayetteville, AR 72701, USA; fmorenad@uark.edu (F.M.); arcabrer@uark.edu (A.R.C.); merosaca@bidmc.harvard.edu (M.E.R.-C.); npgreene@uark.edu (N.P.G.)

**Keywords:** biological sex differences, inflammatory gene expression, protein imbalance, anabolic suppression, muscle wasting

## Abstract

Cancer cachexia (CC) is a multifactorial wasting syndrome characterized by a significant loss in lean and/or fat mass and represents a leading cause of mortality in cancer patients. Nutraceutical treatments have been proposed as a potential treatment strategy to mitigate cachexia-induced muscle wasting. However, contradictory findings warrant further investigation. The purpose of this study was to determine the effects of leucine supplementation on skeletal muscle in male and female *Apc^Min/+^* mice (APC). APC mice and their wild-type (WT) littermates were given normal drinking water or 1.5% leucine-supplemented water (n = 4–10/group/sex). We measured the gene expression of regulators of inflammation, protein balance, and myogenesis. Leucine treatment lowered survival rates, body mass, and muscle mass in males, while in females, it had no effect on body or muscle mass. Leucine treatment altered inflammatory gene expression by lowering *Il1b* 87% in the APC group and decreasing *Tnfa* 92% in both WT and APC males, while it had no effect in females (*p* < 0.05). Leucine had no effect on regulators of protein balance and myogenesis in either sex. We demonstrated that leucine exacerbates moribundity in males and is not sufficient for mitigating muscle or fat loss during CC in either sex in the *Apc^Min/+^* mouse.

## 1. Introduction

Data from the World Health Organization (WHO) show that cancer is a leading cause of death worldwide, and it accounted for approximately 10 million deaths in 2020 [1]. Colorectal cancer is a common form of cancer, with 1.93 million cases and 916,000 deaths reported in 2020 [1,2]. Cancer cachexia (CC) is present in approximately 80% of advanced-cancer patients, worsens patients’ overall outcomes, and increases the risk of mortality; up to 40% of all cancer-related deaths are directly attributed to cachexia [3]. CC is a physiologically complex wasting syndrome, currently clinically defined by a loss of body mass greater than 5% in a 6-month period. This is primarily attributed to a loss in lean mass, which may be accompanied by a loss in fat mass [4,5]. CC is commonly characterized by a state of systemic inflammation and circulating proinflammatory cytokines that are emblematic of this state [6]. There is strong evidence of a correlation between elevated levels of pro-inflammatory cytokines and cachexia; therefore, chronic inflammation is one of the key factors contributing to muscle atrophy in CC [7]. Despite significant efforts toward understanding CC, no effective therapeutic treatments against CC are currently available [8,9,10,11].

A key factor in the characterization of CC is a negative muscle protein balance, favoring catabolism over anabolism. There have been numerous nutritional intervention studies carried out in the hopes of correcting this protein imbalance [12,13,14]. Leucine is a branched-chain amino acid (BCAA) known to directly stimulate the mechanistic target of rapamycin and mTORC, mediating protein synthetic signaling [15]. Currently, leucine is considered a potential therapeutic agent in CC [16]; however, contradictory findings on its effects warrant further investigation. Studies have observed leucine supplementation in rats bearing Walker-256 breast-cancer tumors improved body and muscle mass, improved protein balance, and moderated pro- and anti-inflammatory mediators [17,18,19]. Despite these positive effects of leucine supplementation in rats with breast-cancer-induced cachexia, Lee et al. (2019) [20] suggested that leucine treatment exacerbates the loss of skeletal muscle mass in mice inoculated with Lewis Lung Carcinoma (LLC) tumor cells. Additionally, several studies reported leucine’s potential adverse effects on the cancer tumor itself, where leucine enhanced tumor growth in preclinical models of pancreatic and bladder cancer [21,22,23]. Therefore, more research is required to define the effects of leucine supplementation during CC across different types of cancer in order to determine if leucine supplementation is a potentially efficacious therapeutic agent for mitigating CC.

Along with metabolic dysregulation accompanying CC, emerging evidence indicates that males and females display biological sex differences in the development of cancer and CC, even in response to cancer treatments [24,25,26]. For example, females with CC often experience less severe losses in body mass, muscle mass, and muscle function compared to males, which may be attributable to the anti-inflammatory effects of estrogen [26]. Additionally, female skeletal muscle exhibits a greater mitochondrial quality and less resistance to fatigue [26,27], and our laboratory has demonstrated the significant protection of mitochondrial quality during the early stages of CC in females [28]. Overall, recent studies indicate that males have greater susceptibility to CC-induced muscle loss [29]; however, few studies observed this sexual dimorphism simultaneously [30,31], with fewer still examining potential therapeutic targeting [29,32]. Further evaluation of CC and potential treatments is critical for a complete understanding of mechanistic alterations and the identification of potential therapeutic approaches across the biological sexes.

To our knowledge, leucine supplementation as a nutritional intervention across both sexes in a pre-clinical model of colorectal CC has not yet been studied. Therefore, the purpose of this study was to determine the effects of leucine supplementation on skeletal muscle in a well-known model of colorectal CC and *Apc^Min/+^*, observing the effects in both male and female mice. By supplementing the normal diet with leucine, we hypothesized that male and female mice would exhibit attenuated muscle wasting. We observed that leucine supplementation differentially affects males and females; leucine supplementation exacerbated moribundity in males, without attenuating muscle or fat loss in either sex.

## 2. Methods

### 2.1. Animals and Housing

All animal experiments were approved by the Institutional Animal Care and Use Committee (IACUC) of the University of Arkansas. *Apc^Min/+^* male mice and C57Bl/6J female mice were initially purchased from Jackson Laboratories for colony production (Apc^Min/+^: IMSR_JAX:002020. C57Bl6/J: IMSR_JAX:000664). Animals were kept in a 12:12 light–dark cycle and given access to normal rodent chow. *Apc^Min/+^* males were bred with C57BL6/J female mice as colon cancer is a contraindication for pregnancy. Mice were genotyped with DNA isolated from tail or ear snips. The *Apc^Min/^*^+^ gene was detected via semi-quantitative PCR, using the forward primer (GGG AAG TTT AGA CAG TTC TCG T) and reverse primer (TGT TGG ATG GTA AGC ACT GAG), with an initial denaturation at 95 °C for 2 min followed by 35 cycles of denaturation at 95 °C for 30 s, annealing at 45 °C for 30 s, extension at 72 °C for 30 s, and a final extension at 72 °C for 7 min. Mice were weaned and genotyped at 4 weeks of age. *Apc^Min/+^* and their WT littermates were randomly assigned to the leucine-enriched water (1.5%) experimental group (WTL; male n = 7, female n = 8, or APCL; male n = 7, and female n = 8) or the normal drinking water control group (WTNL; male n = 7, female n = 8, or APCNL; male n = 7, female n = 8) until tissue collection and euthanasia at 20 weeks of age [33,34]. This led to the creation of 4 experimental groups per sex based on genotype and leucine supplementation.

### 2.2. Leucine Supplementation

Mice receiving leucine supplementation received a 1.5 g/100 mL dose of leucine (Fisher Scientific; BP385-100) through their drinking water. Leucine dosage was predetermined based on previous research published by our laboratory, as well as previous research by Li et al. (2012) [34,35]. L-leucine was dissolved in tap water to create a 1.5% leucine solution. The mixture was heated to 70 degrees for 40 min to facilitate the dissolving process. The mixture was cooled down to room temperature before administration to the animals. Food and water intake were measured and recorded weekly; the intake was not different across groups.

### 2.3. Euthanasia and Tissue Collection

Animal tissues, organs, and blood plasma were collected under 3% isoflurane prior to euthanasia at 20 weeks of age or if mice became moribund prior to the collection date. Moribundity was determined in conjunction with the veterinary staff and included these clinical manifestations: hunched posture, lethargy, impaired mobility, severe and rapid weight loss, and no response to external stimuli. Mice that displayed these symptoms were euthanized prior to the planned experimental endpoint. Tissues and organs were weighed and snap-frozen in liquid nitrogen and stored at −80 °C for further processing. The tibialis anterior (TA) muscle was selected for histological and biochemical analysis throughout this study due to the mixed fiber nature and its susceptibility to cachexia [36]. TA muscle was submerged in optimum cutting temperature compound (OCT) and then placed in liquid-nitrogen-cooled isopentane. OCT mounted tissue was then stored at −80 °C for future histological and microscopic analysis.

### 2.4. Histology

TA muscle stored in OCT were cryo-sectioned into 10 μm thick sections on polarized microscope slides. Muscle sections were then histologically stained for succinate dehydrogenase (SDH). Oxidative phenotype was quantified and analyzed by counting relative presence of SDH+ (purple) and SDH- (white) fibers. Following SDH staining, the cross-sectional area (CSA) of muscle was determined by a blinded researcher who manually traced SDH+ and SDH- fibers. Images were analyzed using Nikon NIS Elements BR software package v4.30.

### 2.5. RNA Isolation, cDNA Synthesis, and Quantitative Real-Time PCR

RNA was isolated from the TA muscle. TA muscle was homogenized with Trizol Reagents (Life Technologies, Grand Island, NY, USA), and phenol–chloroform extraction was performed following homogenization. RNA was ethanol-precipitated and diluted in 70% of diethyl pyrocarbonate-treated ethanol. Total RNA was isolated using an Invitrogen PureLink^TM^ RNA Mini Kit (Invitrogen, Waltham, MA, USA; 12183018A). Total RNA was eluted in rNase-free water. RNA concentration and purity were measured using a BioTek Take3 micro-volume microplate with a BioTek Synergy HTX Multi-Mode Microplate Reader, IVD (Fisher Scientific, Waltham, MA, USA; BTS1LASI). A 260/280 nm ratio of >2 was utilized as the quality assessment. Following RNA extraction, samples were stored at −80 °C for future use. RNA was reverse transcribed to cDNA from 1 μg of total RNA using Superscript Vilo cDNA synthesis kit (Invitrogen; 11755-250) in a final volume of 20 μL at 25 °C for 10 min, followed by 42 °C for 50 min, and 70 °C for 15 min. PCR was performed using QuantStudio 3 Real-Time PCR system (Applied Biosystems, Waltham, MA, USA; A28571). A 25 μL reaction composed of TaqMan probes plus Taqman Universal Master Mix (Applied Biosystems; 4305719) was used to amplify cDNA. Samples were incubated at 95 °C for 4 min, followed by 45 cycles of denaturation, annealed, and extended at 95 °C and 60 °C, respectively. TaqMan fluorescence was measured at the end of the extension of each cycle. All targets were assayed using Taqman probes (Applied Biosystems) for the following targets: *18S* (Mm03928990_g1), *Il6* (Mm00446190_m1), *Il1b* (Mm00434228_m1), *Myod* (Mm00440387_m1), *Myogenin* (Mm00446194_m1), *Pax7* (Mm01354484_m1), *Nfkb* (Mm00476361_m1), *Tnfa* (Mm00476361_m1), *Ubc* (Mm01198158_m1), *Deptor* (Mm01195339_m1), *Igf1* (Mm00439560_m1), *Foxo1* (Mm00490671_m1), *Foxo3* (Mm01185722_m1), *Fbox32* (Mm00499523_m1), *Trim63(Murf1)* (Mm01185221_m1), and *Ddit4(Redd1)* (Mm00512504_g1). The RT-qPCR measured cycle threshold (Ct) and the ΔCt value were calculated as the difference between the target Ct value and the *18S* Ct value. The final quantification of mRNA abundance was calculated using the ΔΔCT method. ΔΔCT = [ΔCt (calibrator) − ΔCt(sample)]. Relative quantifications were then calculated as 2^−ΔΔCt^. *18S* Ct values were confirmed to not differ between experimental conditions.

### 2.6. Statistical Analysis

Differences between groups within each biological sex were determined via two-way ANOVA (genotype [WT v APC] by leucine [NL v L]) for a global analysis, where a Student–Newman–Keuls post hoc test was then performed to evaluate differences between means when a significant interaction effect was observed. Any comparison between sexes was performed visually and based upon the contrast of differential or similar statistical effects between sexes. Statistical significance was set at α = 0.05. All data were analyzed, figures were compiled using GraphPad Prism version 9.5.1 (528) (Graphpad Software LLC., Boston, MA, USA), and data were expressed as mean ± standard error of the mean (SEM).

## 3. Results

### 3.1. Leucine Negatively Affects Survival in Male APC^Min/+^ Mice

The male APCL mice showed a 10% decrease in survival rate after 14 weeks, 40% by 19 weeks, and 100% by 20 weeks (Figure 1A). The female APCNL was the only female group to exhibit survival loss prior to the 20-week endpoint (Figure 1B). Body mass was measured over the course of 20 weeks. Male WTNL and WTL groups did not show differences in body mass over time. In male APCNL mice, body mass reduction began to happen at 17 weeks. In male APCL mice, body mass reductions occurred as early as 14 weeks. Additionally, beginning at 15 weeks, male APCL body mass was lower than APCNL (Figure 1C, *p* < 0.05). In females, there were no reductions in body mass between WTNL and WTL. APCNL and APCL mice began to experience losses in body mass after 17 weeks (Figure 1D, *p* < 0.05).

### 3.2. Validation of Cancer Cachectic Phenotype

In males, there was an interaction between genotype and treatment that altered endpoint body mass. There was no difference in between the WTNL and WTL groups; however, male APCNL body mass was 16% lower than WTNL body mass, APCL body mass was 30% lower than WTL, and APCL body mass was 13% lower than APCNL (Table 1 & Figure 1E, *p* < 0.05). In females, endpoint body mass in APC was 14% lower than WT, independent of leucine (Table 1 & Figure 1F, *p* < 0.05). There was a main effect observed in APC male mice, who had 16%, 33%, and 75% lower soleus, plantaris, and TA muscle mass, respectively, compared to the WT group (Table 1, *p* < 0.05). There was a double main effect of genotype and treatment in male gastrocnemius muscle, where muscle mass was 44% lower in APC groups compared to WT, and 9% lower in L compared with NL (Table 1, *p* < 0.05). There was a main effect of the genotype in females, where APC mice had 27%, 39%, 43%, and 75% lower soleus, plantaris, gastrocnemius, and TA muscle mass, respectively, compared to the WT groups (Table 1, *p* < 0.05). In male mice, heart mass was 7% lower in the APC groups compared to WT (Table 1, *p* < 0.05). In contrast, in female mice, heart mass was 11% higher in APC groups compared with WT (Table 1, *p* < 0.05). In both male and female mice, spleen mass was 138% and 124% higher compared with WT groups, respectively (Table 1, *p* < 0.05). Liver mass was unchanged in males; however, in female mice, leucine treatment lowered liver mass 13% in WTL and APCL groups compared with WTNL and APCNL groups (Table 1, *p* < 0.05). Gonadal fat mass in APC males was 176% lower than WT groups (Table 1 & Figure 1G, *p* < 0.05). Importantly, in females, gonadal fat mass was 177% lower in APCNL compared with WT groups, and there was a complete absence of gonadal fat in the APCL group (Table 1 & Figure 1H, *p* < 0.05). Finally, there was no effect of leucine on the total polyp count in either male or female mice (Table 1).

In males, SDH+ CSA was 22% lower in the APC groups compared to WT, due to a main effect of the genotype (Figure 2A, *p* < 0.05). SDH- CSA was 28% lower in APC group compared to WT due to a main effect of genotype (Figure 2C, *p* < 0.05). In females, there was no difference in SDH+ CSA between groups (Figure 2B); however, SDH- CSA was 22% lower in the APC groups compared with WT, due to a main effect of the genotype (Figure 2D, *p* < 0.05). Representative images of muscle cross sections stained for SDH were obtained for both males and females (Figure 2E,F).

### 3.3. Leucine Alters Inflammatory Gene Expression Only in Male Mice

In males, *Il1b* mRNA abundance was 3-fold higher in the APCNL group compared with WTNL, and 87% lower in APCL compared with APCNL (Figure 3A, *p* < 0.05). *Il6* was 2-fold higher in APC groups compared with WT groups due to a main effect of the genotype (Figure 3C, *p* < 0.05). *Tnfa* mRNA abundance was 92% lower in leucine-treated groups (WTL and APCL) compared to non-leucine-treated groups (WTNL and APCNL) due to a main effect of the treatment (Figure 4E, *p* = 0.0535). *Nfkb* mRNA abundance was unchanged across groups, regardless of genotype or treatment (Figure 3G). In females, *Il1b, Il6, Tnfa,* and *Nfkb* mRNA abundances were 6-fold, 4-fold, 2-fold, and 36% higher in the APC, respectively, compared with WT, due to a main effect of genotype (Figure 3B,D,F,H, *p* < 0.05).

### 3.4. Leucine Supplementation Did Not Affect the Induction of Anabolic Suppressor Gene Expression in CC

In males, *Igf1* and *Deptor* mRNA abundance were unchanged across groups (Figure 4A,C). *Redd1* mRNA abundance was 5-fold higher in APC groups compared with WT independent of leucine supplementation (Figure 4E, *p* < 0.05). In females, *Igf1* and *Deptor* mRNA abundance were unchanged (Figure 4B,D). *Redd1* mRNA abundance was 9-fold higher in APC groups compared with WT, independent of leucine supplementation (Figure 4F, *p* < 0.05).

### 3.5. Leucine Supplementation Did Not Affect Induction of Gene Expression in Markers for Protein Degradation in Apc^Min/+^

In males, *Fbxo32, Murf1, Foxo1, Foxo3,* and *Ubc* mRNA abundance were 5-fold, 5-fold, 2-fold, 80%, and 2-fold higher, respectively, in APC compared with WT, due to a main effect of the genotype (Figure 5A,C,E,G,I, *p* < 0.05). In females, *Fbxo32, Murf1, Foxo1, Foxo3,* and *UBC* mRNA abundance were 3-fold, 4-fold, 2-fold, 80%, and 3-fold higher, respectively, in the APC compared with WT, due to a main effect of the genotype (Figure 5B,D,F,H,J, *p* < 0.05).

### 3.6. Biological Sex Differences in Gene Expression of Myogenic Regulators, Independent of Leucine Supplementation

In males, *Pax7* mRNA abundance was 67% lower in the APC group compared with WT, due to a main effect of the genotype (Figure 6A, *p* < 0.05). *Myod* mRNA abundance remained unchanged across groups (Figure 6C). *Myogenin* mRNA abundance was 2-fold higher in APC compared with WT, due to a main effect of genotype (Figure 6E, *p* < 0.05). In females, *Pax7* mRNA abundance is 52% lower in APC compared with WT, due to a main effect of the genotype (Figure 6B, *p* < 0.05). *Myod* and *Myogenin* were increased 2-fold and 82%, respectively, in APC groups compared to WT, due to a main effect of the genotype (Figure 6D,F, *p* < 0.05).

## 4. Discussion

In the present study, we sought to determine if leucine supplementation would attenuate CC in a common preclinical model of colorectal cancer and to observe whether this would be the effect for both sexes. We hypothesized leucine supplementation would improve survival outcomes and attenuate muscle wasting in male and female *Apc^Min/+^* mice, given that leucine is a branched-chain amino acid known for its direct stimulation of mTORC-mediated protein synthesis. Contrary to our hypothesis, male *Apc^Min/+^* mice provided with leucine supplementation had worse survival outcomes, exacerbated body mass loss, and lowered gastrocnemius muscle mass. In females, leucine did not impact survival, body mass, or muscle masses; however, in contrast with the male mice, we observed a complete absence of gonadal fat. In addition to effects on phenotypic characteristics, we observed leucine-attenuated aspects of inflammatory gene expression in *Apc^Min/+^* male skeletal muscle but not in females. Therefore, we demonstrated biological sex differences in the response to leucine treatment in *Apc^Min/+^* mice. Most importantly, leucine did not have a positive effect on the phenotypic characteristics of cachexia and demonstrated deleterious effects on tumor-bearing mice. Therefore, based on our data, leucine supplementation does not appear to be an efficacious therapeutic strategy in CC.

In this study, we demonstrate the existence of biological sex differences in survivability and other phenotypic characteristics, such as muscle and fat mass, in response to leucine treatment. Notably, leucine appears to have a negative effect on male survivability, given that our APCL mice became moribund before their 20-week endpoint, and body mass began to steadily decline at 15 weeks of age, while APCNL male mice did not become moribund until 19 weeks of age, and body mass loss began at 18 weeks. In contrast, in female mice, only APCNL mice became moribund prior to the planned experimental endpoint of 20 weeks, with all APCL female mice making it to their 20-week endpoint. Importantly, both female APCNL and APCL groups became cachectic; therefore, leucine did not improve survival outcomes in females, rather, leucine may have had less detrimental effects on survival outcomes in females compared with males. Furthermore, leucine had a main effect on lowering heart mass in *Apc^Min/+^* males, indicating the presence of cardiac cachexia. This potential cardiac cachexia could be a driver of the increased moribundity seen in male *Apc^Min/+^* treated with leucine, and this should be examined further in future studies. Together, these results indicate that leucine worsened overall outcomes specifically in male mice. Interestingly, female APCL exhibited a complete absence of gonadal fat. Leucine has well-documented effects on lipid metabolism [37,38], which may be dysregulated in a cachectic environment. Despite this complete loss of fat mass, female APCL mice did not experience the same severity of body mass loss that APCL male mice did. Several studies have demonstrated biological sex differences in these phenotypic characteristics between males and females. Clinical studies have observed worsened survival outcomes in males with cachexia and found that more males presented diminished muscle mass compared with females [39,40]. A study observing *Apc^Min/+^*-induced cachexia reported diminished body mass in males and females; however, males experienced greater reductions in body mass than females [41]. Our results corroborate these prior findings, and demonstrate that leucine has detrimental effects rather than beneficial effects on survivability, body mass loss, and muscle mass loss in males, and fat mass loss in females.

The effect of leucine on tumors has previously been examined and could provide more evidence for the effects noted in the current study. It was demonstrated that leucine supplementation in pancreatic cancer exacerbated tumor growth and aggressivity [21]. Interestingly, one study observing leucine deprivation in breast cancer found that leucine deprivation inhibited tumor proliferation and promoted the apoptosis of tumor cells [42]. Unfortunately, in the present study, the effect of leucine supplementation on tumor characteristics was not measured outside of total polyp count, which remained unchanged following leucine supplementation. There may have been an effect of leucine on other factors, such as tumor progression or tumor volume in males, which was not present in females, and this may have been a contributing factor in the exacerbated moribundity displayed by male *Apc^Min/+^* mice.

Male and female *Apc^Min/+^* mice exhibited splenomegaly, regardless of leucine supplementation. Splenomegaly is a hallmark indicator of systemic inflammation [37]. Furthermore, pro-inflammatory cytokine *Il6* is upregulated in skeletal muscle in the *Apc^Min/+^* model of CC [41,43,44,45]. Prior studies established biological sex differences in *Il6* expression in *Apc^Min/+^* mice and found that cachexia in *Apc^Min/+^* mice is Il6-dependent in male mice but not in female mice. We observed differences in pro-inflammatory muscle gene expression between the males and females, whereby females exhibited increased levels of *Il6*, *Il1b*, *Tnfa*, and *Nfkb* in the cachectic *Apc^Min/+^* mice, independent of leucine treatment. In *Apc^Min/+^* males, however, leucine attenuated the increase in *Il1b*. Additionally, leucine lowered *Tnfa* levels in both WT and APC groups compared with the groups given normal drinking water. *Il1b* is known to stimulate the production of *Il6* [46], and the increased expression of *Il1b* is related to CC [47]. Therefore, leucine’s ability to attenuate an increase in *Il1b* in male cachectic *Apc^Min/+^* mice could be beneficial via reducing downstream expression of *Il6* [48]. However, our data show that *Il6* in leucine-treated males remained elevated. Therefore, our data suggest that the mechanism for enhanced expression of *Il6* in CC is most likely not only dependent on *Il1b* expression. Our data corroborate findings of prior studies suggesting that cachexia in male Apc^Min/+^ is Il6-dependent [43]. However, our observations revealed that the administration of leucin led to a reduction in the expression of inflammatory markers 9Il1b and tnfa), except for Il6, which could potentially be correlated with the heightened moribundity observed in ApcMin/+ males. Conversely, leucine did not impact the induction of inflammatory cytokines in the muscles of female mice, reflective of a biological-sex-specific effect of leucine. It is important that leucine’s effect on immune cell infiltration in male and female cancer cachexia is considered as an avenue for future research. There is a growing body of evidence supporting the importance of immune cell regulation in cancer cachexia [49,50]; therefore, these factors should be studied more thoroughly.

Despite the effect of leucine on lowering inflammatory muscle gene expression in males, our data show that leucine had no effect on improving the markers for the protein imbalance that occurs in cachexia. Leucine supplementation did not stimulate the common promotor of protein synthesis, *Igf1*, nor did it significantly mitigate the induction of the repressor of protein synthesis, *Redd1*. Intriguingly, *Deptor*, another repressor of protein synthesis, remained unchanged across groups in both males and females, corroborating recent research from our laboratory, where we found an elevation of *Redd1* but not *Deptor* in the C26 model of colorectal cancer cachexia [31]. Prior research from our group found an elevation of Deptor protein content in an LLC model of cachexia [9,28]. Our combined data now strongly suggest a specific reliance on *Redd1* to mediate repression of muscle protein synthesis during colorectal-cancer-induced cachexia. Furthermore, a panel of five genes related to protein breakdown were all increased in the *Apc^Min/+^* groups of both males and females, with no overt impact of leucine on either attenuating or exacerbating this induction. In males, it is possible that this elevation may be due to *Il6*, given that persistent *Il6* expression is a factor linked to enhanced protein degradation in CC [51]. However, in females, the mechanism is less clear. *Il6*, *Il1b*, *Tnfa*, and *Nfkb* were all elevated in the muscles of females; therefore, considering prior literature [41,43], if pro-inflammatory cytokines are impacting protein breakdown, they are likely not *Il6*-dependent.

Previous research from our laboratory has shown altered myogenic response during CC in male and female mice inoculated with LLC tumors [9,28]. Additional research from He et al. (2013) [52] discovered that *Pax7* is elevated in skeletal muscle in the early stages of CC in the Colon-26 (C26) model of colorectal cancer cachexia. We observed mRNA abundance levels of myogenic regulators—*Pax7*, *Myod*, and *Myogenin* in male and female *Apc^Min/+^* mice. We found lowered *Pax7* mRNA abundance in males and females, regardless of leucine supplementation, corroborating prior findings of our laboratory, which show decreased *Pax7* mRNA levels in tumor-bearing male and female LLC mice [9,28]. Furthermore, we found an elevated mRNA abundance of *Myod* in females, but not in males, which does not corroborate prior data. Brown et al. (2018) [9] found decreased *Myod* mRNA levels in male tumor-bearing mice, and Lim et al. (2022) [28] found no change in *Myod mRNA* levels in female tumor-bearing mice. Furthermore, we found increases in *Myogenin* mRNA in both sexes. Intriguingly, Moresi et al. (2010) [53] observed a linkage between myogenin expression and upregulation in E3 ligases, *Fbxo32* and *Murf1*, in a mouse model of neurogenic atrophy. In our model, we observed elevations in E3 ligases, corroborating this suggested mechanism and potentially extending it into cachexia. Taken together, there are clear biological sex differences that exist in the myogenic response of cancer cachexia, and these responses may be dependent on the stage and progression of cancer, as well as the type of cancer that induces cachexia.

In conclusion, the purpose of this study was to observe the effect of leucine supplementation, a proposed therapeutic, on skeletal muscle wasting in CC. We hypothesized that leucine supplementation would attenuate wasting via the stimulation of protein synthesis. However, we observed that leucine provided no beneficial effects to males or females, and in fact exacerbated moribundity in *Apc^Min/+^* males. Regardless of biological sex, leucine supplementation appears to provide no benefit as a therapeutic in CC. The biological sex differences observed in response to leucine provide further insights into the expanding body of knowledge regarding sex differences in cachexia and may extend our understanding of mechanisms. Additionally, further research into potentially deleterious effects may be warranted due to current opportunities in clinical practice and potential mechanistic insights.

## Figures and Tables

**Figure 1 nutrients-15-04570-f001:**
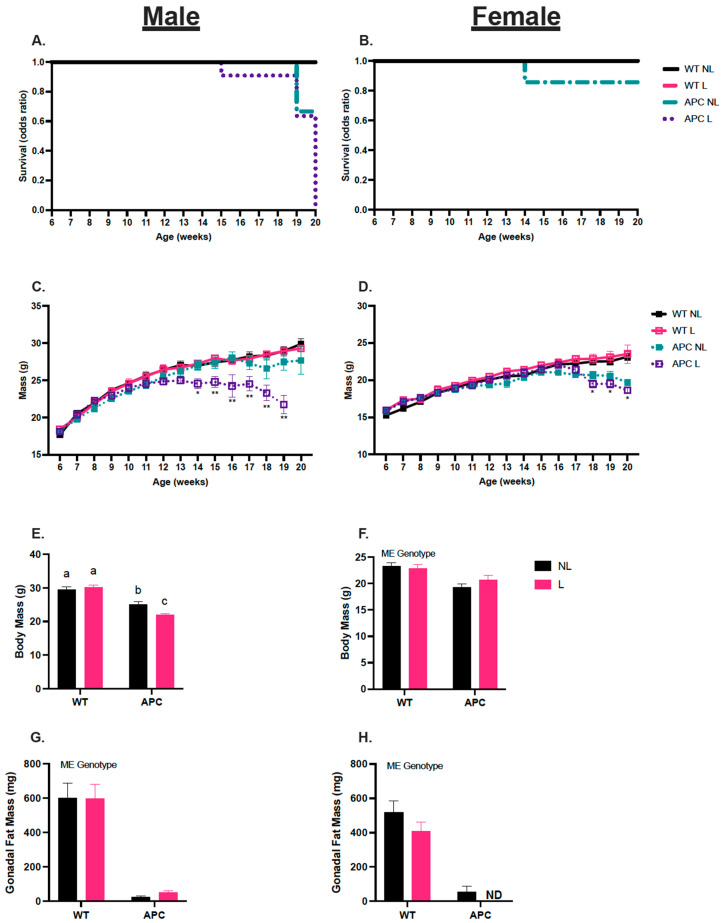
Leucine negatively affects survival in male *APC^Min/+^* mice. (**A**,**B**) Survival rate over time for males and females. Survivability was determined when animals became moribund, and euthanasia was carried out. (**C**) Body mass (g) over time (weeks) for male mice. * APC group lower than WT. ** APC group lower than WT, and APCL lower than APCNL (*p* < 0.05). (**D**) Body mass (g) over time (weeks) for female mice. * main effect of genotype at individual data point (*p* < 0.05). (**E**,**F**) End-point body mass (g) for males and females. (**G**,**H**) Gonadal fat mass (mg) for males and females, respectively. ND indicates “No Data”. Data expressed as mean ± SEM. Letters (a, b, c) indicate differences between groups; the interaction effect considering Student–Newman–Keuls with adjusted *p* < 0.05. ‘ME Genotype’ indicates the main effect of genotype (WTNL vs. APCNL) with α = 0.05. A n of 3–17 was used for all groups.

**Figure 2 nutrients-15-04570-f002:**
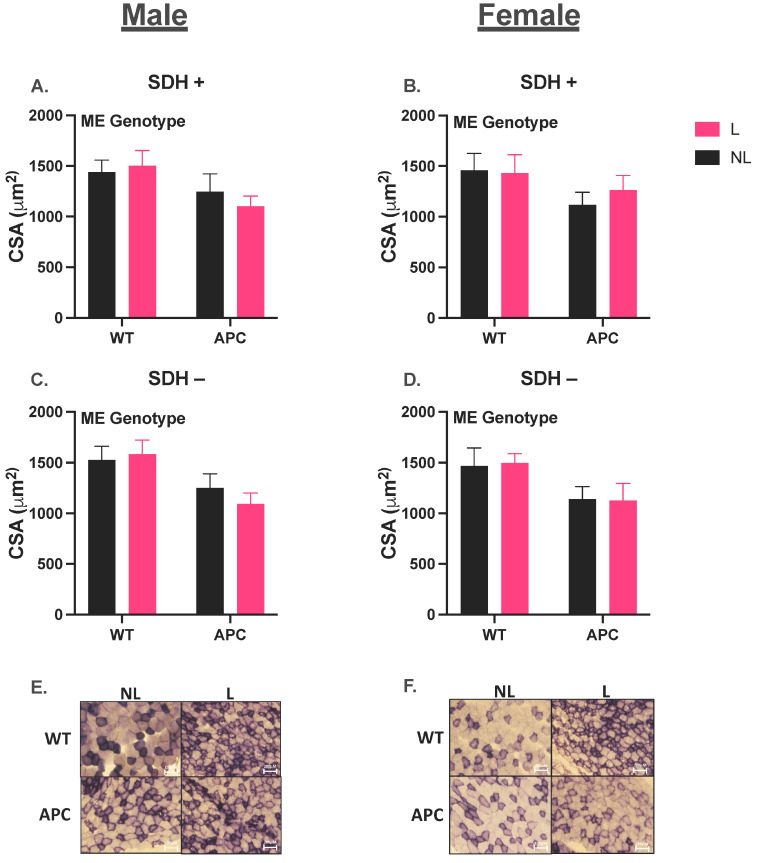
Histological assessment of oxidative capacity. (**A**,**B**) Cross-sectional area of male and female SDH+ TA muscle fibers. (**C**,**D**) Cross-sectional area of male and female SDH− fibers. (**E**,**F**) Representative images of male and female muscle cross sections stained for SDH. Data are expressed as mean ± SEM. ‘ME Genotype’ denotes a main effect of genotype (WT vs. APC) with α = 0.05. An n of 6 was used for all groups.

**Figure 3 nutrients-15-04570-f003:**
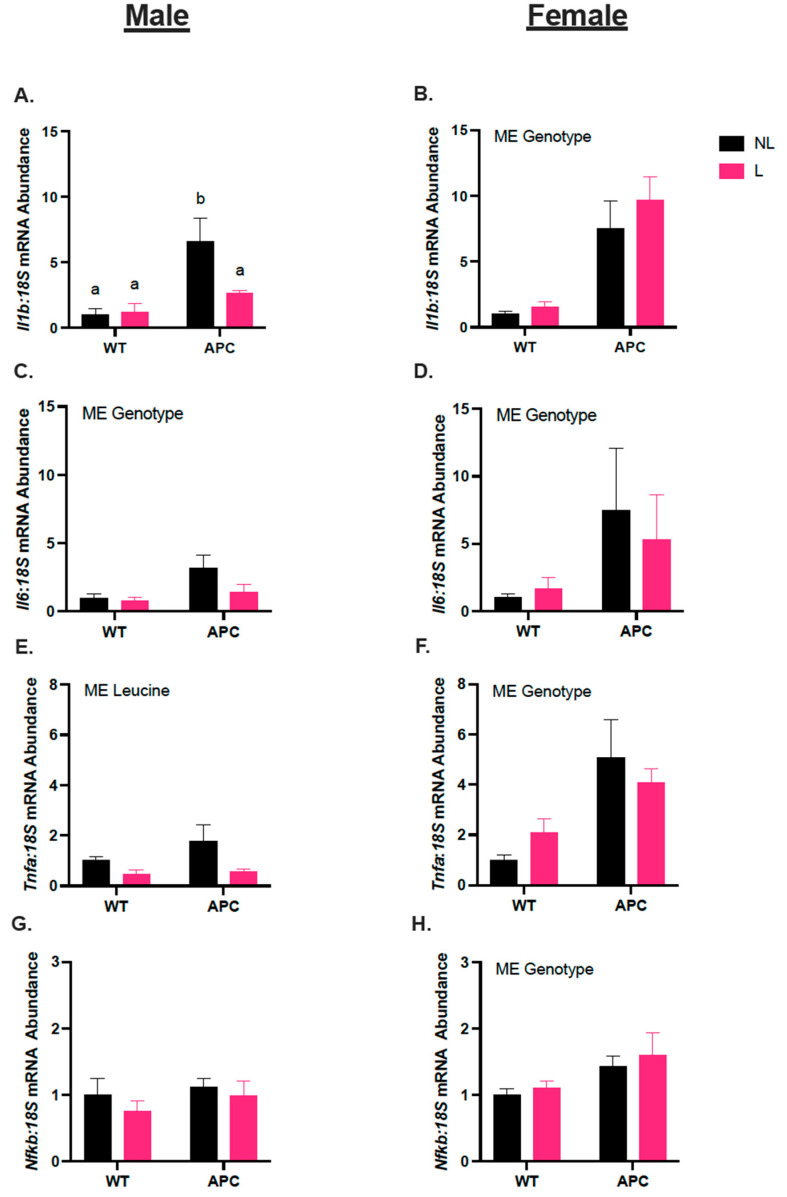
Leucine alters inflammatory gene expression only in male mice. (**A**,**C**,**E**,**G**) Male *Il-1b*, *Il-6*, *Tnf-a*, and *Nf-kb* mRNA abundance. (**B**,**D**,**F**,**H**) Female *Il-1b*, *Il-6, Tnf-a*, and *Nf-kb* mRNA abundance. Data are shown as mean ± SEM. Letters (a, b) indicate differences between groups, interaction effect considering Student–Newman–Keuls with adjusted *p* < 0.05. “ME genotype” indicates main effect genotype (WT vs. APC), ‘ME treatment’ indicates a main effect of leucine treatment (WTNL and APCNL vs. WTL and APCL) with α = 0.05. An n of 4–10 was used for all groups.

**Figure 4 nutrients-15-04570-f004:**
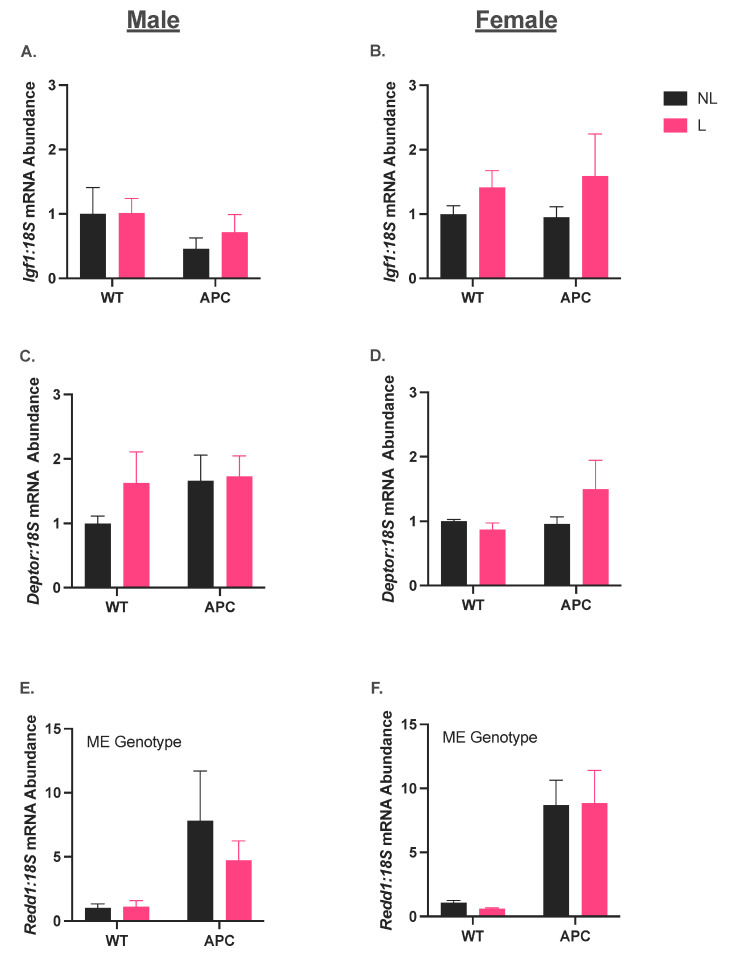
Leucine supplementation did not affect the induction of anabolic suppressor gene expression in CC. (**A**,**C**,**E**) Male mRNA abundance for *Igf1*, *Deptor*, and *Redd1.* (**B**,**D**,**F**) Female mRNA abundance for *Igf-1*, *Deptor*, and *Redd1*. Data are shown in mean ± SEM. ‘ME genotype’ denotes a main effect of genotype (WT vs. APC) with α = 0.05. An n of 5–10 was used for all groups.

**Figure 5 nutrients-15-04570-f005:**
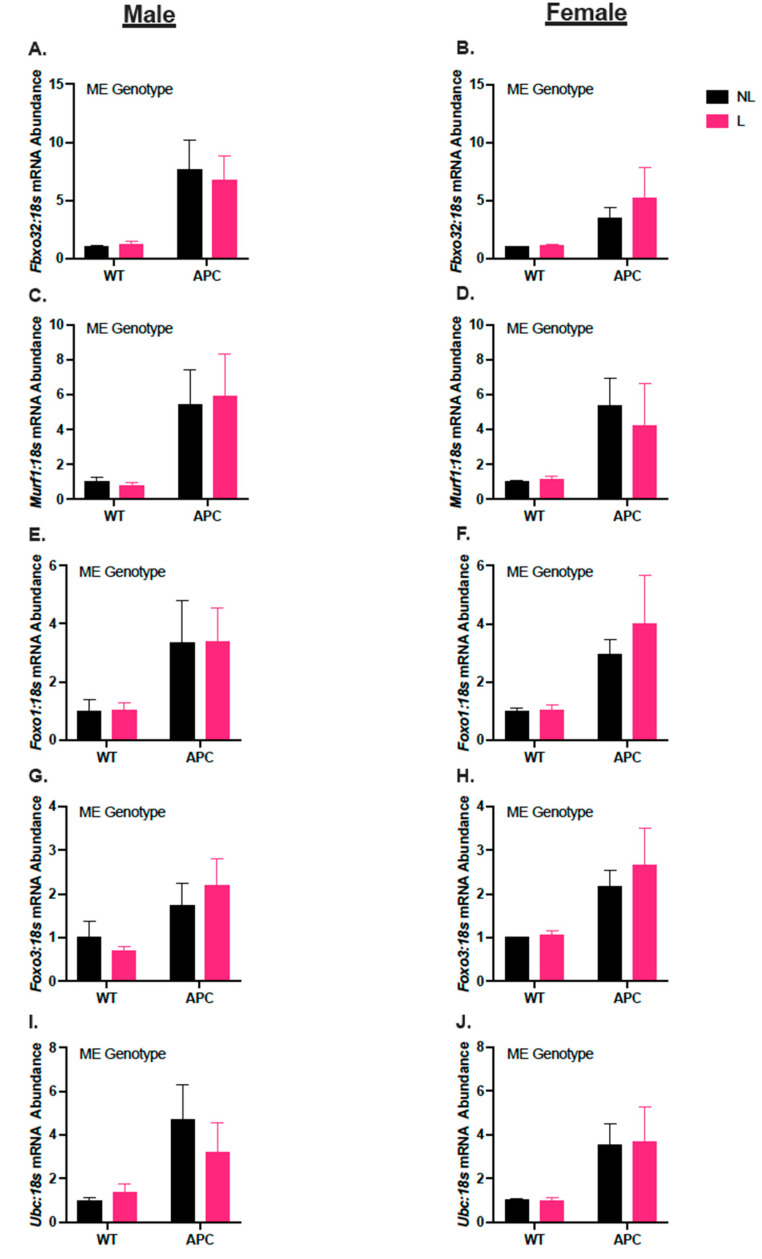
Leucine supplementation did not affect the induction of gene expression in markers for protein degradation in *Apc^Min/^*^+^. (**A**,**C**,**E**,**G**,**I**) Male mRNA abundance of *Fbxo32*, *Murf-1*, *Foxo1*, *Foxo3*, and *Ubc,* respectively. (**B**,**D**,**F**,**H**,**J**) Female mRNA abundance of *Fbxo32*, *Murf-1*, *Foxo1*, *Foxo3*, and *Ubc,* respectively. Data expressed as mean ± SEM. ME genotype denotes a main effect of the genotype (WT vs. APC) with α = 0.05. An n of 4–10 was used for all groups.

**Figure 6 nutrients-15-04570-f006:**
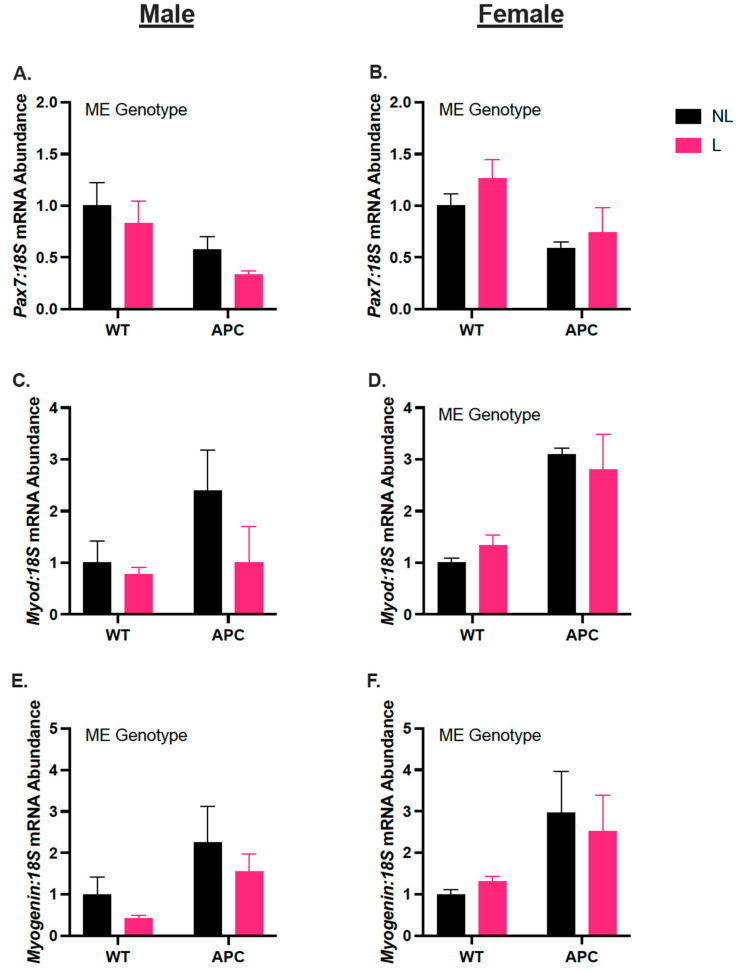
Biological sex differences in gene expression myogenic regulators, independent of leucine supplementation. Male (**A**,**C**,**E**) mRNA abundance of *Pax7*, *Myod*, and *Myogenin,* respectively. (**B**,**D**,**F**) Female mRNA abundance of *Pax7*, *Myod*, and *Myogenin,* respectively. Data are expressed as mean ± SEM. The ME genotype denotes a main effect of the genotype (WT vs. APC) with α = 0.05. An n of 4–10 was used for all groups.

**Table 1 nutrients-15-04570-t001:** Phenotypic data.

**Males**	**WTNL**	**WTL**	**APCNL**	**APCL**
$ Body Mass (g)	29.5 ± 0.8 ^a^	29.7 ± 0.8 ^a^	25.0 ± 0.8 ^b^	21.9 ± 0.5 ^c^
* Soleus (mg/mm)	0.59 ± 0.04	0.56 ± 0.03	0.47 ± 0.02	0.49 ± 0.03
* Plantaris (mg/mm)	1.08 ± 0.04	1.11 ± 0.02	0.84 ± 0.04	0.74 ± 0.05
*# Gastrocnemius (mg/mm)	8.54 ± 0.20	7.99 ± 0.30	5.62 ± 0.30	4.92 ± 0.30
* Tibialis Anterior (mg/mm)	3.00 ± 0.07	3.25 ± 0.06	1.98 ± 0.20	1.94 ± 0.08
* Heart (mg)	131.6 ± 4.50	130.9 ± 3.10	128.2 ± 5.61	116.1 ± 3.62
* Spleen (mg)	90.1 ± 2.32	97.1 ± 2.15	515.7 ± 39.30	492.5 ± 39.40
Liver (mg)	1364.30 ± 46.50	1318.1 ± 34.30	1306.3 ± 142.50	1185.4 ± 77.90
* Gonadal Fat (mg)	601.25 ± 85.8	596.13 ± 84.30	25.47 ± 6.80	49.57 ± 12.30
Total Polyp count	ND	ND	46 ± 4	34 ± 5
**Females**	**WTNL**	**WTL**	**APCNL**	**APCL**
* Body Mass (g)	23.2 ± 0.7	22.9 ± 0.7	19.3 ± 0.6	19.3 ± 0.9
* Soleus (mg/mm)	0.57 ± 0.04	0.39 ± 0.04	0.39 ± 0.02	0.44 ± 0.04
* Plantaris (mg/mm)	0.87 ± 0.05	0.86 ± 0.04	0.58 ± 0.04	0.59 ± 0.04
* Gastrocnemius (mg/mm)	6.44 ± 0.10	6.39 ± 0.10	4.24 ± 0.30	4.20 ± 0.30
* Tibialis Anterior (mg/mm)	2.49 ± 0.10	2.28 ± 0.09	1.59 ± 0.09	1.48 ± 0.10
* Heart (mg)	104.1 ± 4.30	109.60 ± 50	114.40 ± 5.50	123.90 ± 5.70
* Spleen (mg)	89.1 ± 4.50	85.70 ± 4.20	378.7 ± 39.30	346.00 ± 48.10
# Liver (mg)	953.9 ± 18.10	880.10 ± 39.10	1011.00 ± 43.90	851.20 ± 97.00
* Gonadal Fat (mg)	516.8 ± 68.20	409.63 ± 53.10	97.83 ± 32.10	ND
Total Polyp count	ND	ND	29.8 ± 4	38.5 ± 16

Data shown as mean ± SEM. Asterisk (*) denotes the main effect of genotype (WT vs. APC) and pound (#) denotes the main effect of leucine (WTNL vs. WTL, APCNL vs. APCL) with significance set at *p* < 0.05. Dollar sign ($) denotes interaction effect considering a Newman–Keuls method with adjusted *p* < 0.05. Superscripts of letters (a, b, c) indicate differences between groups. ND indicates “No Detected”. An n of 7–12 was used for all groups.

## Data Availability

The datasets used and analyzed in the current study are available from the corresponding author on reasonable request.

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
