# Peer review of "Leucine Supplementation Exacerbates Morbidity in Male but Not Female Mice with Colorectal Cancer-Induced Cachexia"

_nutrients, 2023, doi:10.3390/nu15214570_

Round 1

Reviewer 1 Report

I am honored to have the opportunity to review this manuscript titled "Leucine supplementation exacerbates morbidity in male, but not female mice, with colorectal cancer-induced cachexia".

In this manuscript, the authors demonstrated that leucine treatment has different effects on the development of cancer cachexia in male and female mice, exacerbating morbidity in male mice.

-The main point I want to address to the authors is how does leucine exacerbate morbidity in male mice? As leucine treatment does not increase gene expression of inflammatory cytokines, it does not alter the diameter of muscle fibers and gonadal fat mass. I believe this is something the authors could add in the discussion section.

-In contrast, Apcmin/+ in female mice which were not treated with leucine exhibited losses prior the planned experimental endpoint of 20 weeks. So, can I say here that leucine treatment increases the lifespan of female mice with cancer-cachexia, even decreasing the gonadal fat mass?

Minor points:

-Leucine treatment was done using. 1.5g/100 mL in drinking water. How do the authors find the ideal concentration of leucine to treat these mice?Please, if it is possible, add the explanation in materials and methods.

-Survivability was determined when the animal became moribund. Please describe what factor the authors used to identify when the mouse was moribund, for example, the mouse stopped eating, became lethargic, or lost 5% of its body mass. Please, if possible, add it in to the methods.

- In figure 2F, the representative image used for SDH staining appears  that leucine treatment increases the CSA of SDH+ and SDH- fibers in female mice. I suggest to the authors to replace the representative image, if possible.

Reviewer 2 Report

In this manuscript, Schrems et al. conducted research to explore whether leucine supplement could prevent colorectal cancer-induced cachexia in a mouse model. The experimental design of the animal model is reasonable, and authors observed the body and muscle mass and further studied gene expression related to inflammation, anabolic suppression, protein degradation, and myogenesis. The authors propose that leucine may not have any beneficial effects on cachexia. Overall, the manuscript is well-written, with minimal errors.

The following are some detailed suggestions that can improve the manuscript.

1.     It is interesting to know whether leucine supplements affect cancer development. The author only showed polyp count. Whether there are any changes in total tumor volume? And stage of the tumor?

2.     In Figure 2, it seems that in E and F, leucine increased SDH+ TA. But this was not reflected in panel A-D.

3.     It is unclear why the authors studied inflammation in the muscle. Is there any immune cell infiltration?

Round 2

Reviewer 1 Report

I want to thank the authors for responding promptly to my questions.

I believe the manuscript is ready for acceptance.

Reviewer 2 Report

All the questions were properly addressed?